# A Real-World Comparative Analysis of Atezolizumab Plus Bevacizumab and Transarterial Chemoembolization Plus Radiotherapy in Hepatocellular Carcinoma Patients with Portal Vein Tumor Thrombosis

**DOI:** 10.3390/cancers15174423

**Published:** 2023-09-04

**Authors:** Soon Kyu Lee, Jung Hyun Kwon, Sung Won Lee, Hae Lim Lee, Hee Yeon Kim, Chang Wook Kim, Do Seon Song, U Im Chang, Jin Mo Yang, Soon Woo Nam, Seok-Hwan Kim, Myeong Jun Song, Ji Hoon Kim, Ahlim Lee, Hyun Yang, Si Hyun Bae, Ji Won Han, Heechul Nam, Pil Soo Sung, Jeong Won Jang, Jong Young Choi, Seung Kew Yoon, Dong Jae Shim, Doyoung Kim, Myungsoo Kim

**Affiliations:** 1Department of Internal Medicine, Incheon St. Mary’s Hospital, College of Medicine, The Catholic University of Korea, Seoul 06591, Republic of Korea; blackiqq@gmail.com (S.K.L.); drswnam@hanmail.net (S.W.N.); 2The Catholic University Liver Research Center, College of Medicine, The Catholic University of Korea, Seoul 06591, Republic of Korea; swleehepa@gmail.com (S.W.L.); atom069@naver.com (H.L.L.); hee82@catholic.ac.kr (H.Y.K.); cwkim@catholic.ac.kr (C.W.K.); dsman@catholic.ac.kr (D.S.S.); cui70@catholic.ac.kr (U.I.C.); jmyangdr@catholic.ac.kr (J.M.Y.); shawnk80@naver.com (S.-H.K.); mjsong95@catholic.ac.kr (M.J.S.); jihoon23@catholic.ac.kr (J.H.K.); ahlim84@hanmail.net (A.L.); oneggu@naver.com (H.Y.); baesh@catholic.ac.kr (S.H.B.); tmznjf@catholic.ac.kr (J.W.H.); hcnam128@catholic.ac.kr (H.N.); pssung@catholic.ac.kr (P.S.S.); garden@catholic.ac.kr (J.W.J.); jychoi@catholic.ac.kr (J.Y.C.); yoonsk@catholic.ac.kr (S.K.Y.); 3Department of Internal Medicine, Bucheon St. Mary’s Hospital, College of Medicine, The Catholic University of Korea, Seoul 06591, Republic of Korea; 4Department of Internal Medicine, Uijeongbu St. Mary’s Hospital, College of Medicine, The Catholic University of Korea, Seoul 06591, Republic of Korea; 5Department of Internal Medicine, St. Vincent’s Hospital, College of Medicine, The Catholic University of Korea, Seoul 06591, Republic of Korea; 6Department of Internal Medicine, Daejeon St. Mary’s Hospital, College of Medicine, The Catholic University of Korea, Seoul 06591, Republic of Korea; 7Department of Internal Medicine, Eunpyeong St. Mary’s Hospital, College of Medicine, The Catholic University of Korea, Seoul 06591, Republic of Korea; 8Department of Internal Medicine, Seoul St. Mary’s Hospital, College of Medicine, The Catholic University of Korea, Seoul 06591, Republic of Korea; 9Department of Radiology, Incheon St. Mary’s Hospital, College of Medicine, The Catholic University of Korea, Seoul 06591, Republic of Korea; 10Department of Radiation Oncology, Incheon St. Mary’s Hospital, College of Medicine, The Catholic University of Korea, Seoul 06591, Republic of Korea

**Keywords:** hepatocellular carcinoma, portal vein thrombosis, survival, progression free survival, response, immune checkpoint inhibitors, transarterial chemoembolization, radiotherapy

## Abstract

**Simple Summary:**

This multicenter cohort study is the first to compare the clinical outcomes between the Atezolizumab-plus-bevacizumab (Ate/Bev) and transarterial-chemoembolization-plus-radiotherapy (TACE + RT) therapies in hepatocellular carcinoma (HCC) patients with portal vein tumor thrombosis (PVTT) who had no metastasis. Through detailed analyses, our study revealed that the Ate/Bev treatment provided superior one-year survival compared to the TACE + RT treatment. The superior outcome of the Ate/Bev therapy was constantly observed in patients with an extensive HCC burden. Meanwhile, patients with unilobar disease demonstrated comparable outcomes between the two treatment groups. Finally, in the propensity score-matching analysis, both one-year survival and progression-free survival rates were higher in the Ate/Bev treatment group. These results suggest that Ate/Bev treatment should be considered as the primary treatment option for HCC patients with PVTT. With respect to TACE + RT, this could also be considered as an alternative treatment option alongside Ate/Bev therapy in patients with unilobar intrahepatic HCC.

**Abstract:**

This study aimed to compare the treatment outcomes of atezolizumab-plus-bevacizumab (Ate/Bev) therapy with those of transarterial chemoembolization plus radiotherapy (TACE + RT) in hepatocellular carcinoma (HCC) patients with portal vein tumor thrombosis (PVTT) and without metastasis. Between June 2016 and October 2022, we consecutively enrolled 855 HCC patients with PVTT. After excluding 758 patients, 97 patients (n = 37 in the Ate/Bev group; n = 60 in the TACE + RT group) were analyzed. The two groups showed no significant differences in baseline characteristics and had similar objective response and disease control rates. However, the Ate/Bev group showed a significantly higher one-year survival rate (*p* = 0.041) compared to the TACE + RT group, which was constantly displayed in patients with extensive HCC burden. Meanwhile, the clinical outcomes were comparable between the two groups in patients with unilobar intrahepatic HCC. In Cox-regression analysis, Ate/Bev treatment emerged as a significant factor for better one-year survival (*p* = 0.049). Finally, in propensity-score matching, the Ate/Bev group demonstrated a better one-year survival (*p* = 0.02) and PFS (*p* = 0.01) than the TACE + RT group. In conclusion, Ate/Bev treatment demonstrated superior clinical outcomes compared to TACE + RT treatment in HCC patients with PVTT. Meanwhile, in patients with unilobar intrahepatic HCC, TACE + RT could also be considered as an alternative treatment option alongside Ate/Bev therapy.

## 1. Introduction

Liver cancer is the seventh most common cancer worldwide, and hepatocellular carcinoma (HCC) is the most dominant type of liver cancer, accounting for the fourth most common cause of cancer mortality [1,2]. Although surveillance for HCC has been conducted using ultrasonography and serum alpha-fetoprotein, more than half of patients with HCC are at an intermediate stage or advanced stage [2]. For patients with advanced-stage HCC, systemic therapy is the mainstream treatment, and sorafenib, a multi-kinase inhibitor, has been widely used as the first-line therapy over the past decade [3].

Recently, immune checkpoint inhibitors (ICIs) have revolutionized the treatment strategy for advanced hepatocellular carcinoma. Particularly, atezolizumab-plus-bevacizumab (Ate/Bev) therapy provides better treatment outcomes compared to sorafenib [4,5]. The IMbrave150 study demonstrated a one-year survival rate of 67.2% with Ate/Bev compared to 54.6% with sorafenib [4]. Therefore, Ate/Bev therapy is recommended as the first-line therapy for advanced HCC, including for patients with portal vein tumor thrombosis (PVTT) [6,7,8,9].

Meanwhile, for HCC patients with macrovascular invasion, including PVTT, transarterial chemoembolization plus radiotherapy (TACE + RT) has also demonstrated better survival than sorafenib in a randomized clinical trial (RCT) [10]. The RCT showed that the TACE + RT group had a significantly longer progression-free survival (PFS) and overall survival compared to the sorafenib group (55.0 vs. 43.0 weeks, respectively; *p* = 0.04) [10]. Based on these results, the guidelines from the 2022 Korean HCC guideline have placed TACE + RT (with a grade B for the quality of evidence) as a recommended treatment option for HCC patients with PVTT. This recommendation stands alongside Ate/Bev therapy, which holds a grade A for the quality of evidence [11].

However, there have been limited data on comparing the efficacy and safety between Ate/Bev and TACE + RT in HCC patients with PVTT. This raises the question of which treatment provides better outcomes for HCC patients with PVTT. To address this question, we aimed to compare the treatment outcomes of Ate/Bev and TACE + RT in HCC patients with PVTT.

## 2. Materials and Methods

### 2.1. Patients

In this retrospective, multicenter study conducted across eight university hospitals that are part of The Catholic University of Korea, HCC patients with PVTT (n = 855), who were treated between June 2016 and October 2022, were consecutively screened for eligibility. Among them, 758 patients were excluded for the following reasons: patients treated with tyrosine kinase inhibitors or hepatic artery infusion therapy (n = 256), patients treated with ICI and RT simultaneously (n = 12), patients with distant metastasis (n = 471), and patients lost to follow-up before three months (n = 19). Finally, 97 patients treated with Ate/Bev (n = 37) or TACE + RT (n = 60) were included and analyzed (Figure 1). HCC was diagnosed based on histologic and/or radiologic findings according to international guidelines [12,13]. The presence of PVTT was confirmed using multiphase contrast-enhanced computed tomography (CT) and/or magnetic resonance imaging (MRI), and the grade of PVTT was classified according to the Liver Cancer Study Group of Japan [12,13,14]. This study was conducted following the Declaration of Helsinki and was approved by the Institutional Review Board of the Catholic University of Korea (XC23RIDI0050).

### 2.2. Treatment Regimens

Ate/Bev therapy was administered intravenously at a dose of 1200 mg of atezolizumab plus 15 mg/kg of body weight of bevacizumab every 3 weeks and continued until the presence of disease progression or intolerable toxicity [4]. For TACE treatment, 2 mg/kg of cisplatin or doxorubicin (50 mg) was infused after selection of the feeding artery for HCC. Subsequently, the feeding arteries were embolized using a mixture of 5–10 mL of cisplatin and iodized oil (Lipiodol Ultra-Fluide; Laboratoire Guerbet, Aulnay-Sous-Bois, France), followed by a gelatin sponge (Gelfoam; Upjohn, Hastings, MI, USA) in patients with appropriate portal blood flow. However, for patients with severely impaired portal blood flow, the embolization was not omitted. TACE treatment was repeated every 6–8 weeks according to the patient’s status [10]. After the first TACE, external beam RT was performed on the PVTT with a planned total dose of 50 Gy in 5 fractions. 

### 2.3. Assessment of Outcomes

Given the recent widespread use of Ate/Bev treatment in Korea, we chose one-year survival as the primary outcome. This was defined as the time interval from the initiation of treatment to either death or the last follow-up of up to one year. Secondary outcomes included the one-year PFS, objective response rate (ORR), and disease control rate (DCR). Treatment responses were assessed by multiphase liver CT or MRI every 6 to 9 weeks according to Response Evaluation Criteria in Solid Tumors 1.1 (RECIST 1.1) [15]. Measurements of the tumor and responses to the treatments were independently reviewed by two expert radiologists (D.J. Shim and D. Kim) without knowledge of the treatment methods and clinical outcomes. PFS was defined as the length of time from the first treatment time to progression according to RECIST 1.1, death, or the last follow-up. Treatment-related adverse events (AEs) were evaluated using the Common Terminology Criteria for Adverse Events version 5.0 (CTCAE v5.0) [16]. 

### 2.4. Statistical Analysis

Baseline characteristics of the included patients are represented as a mean ± standard deviation or median (interquartile range) for quantitative variables and as counts (%) for categorical variables, as appropriate. Comparisons between groups were analyzed using the Student’s *t*-test or Mann–Whitney U test for continuous variables and the chi-square test or Fisher’s exact test for categorical variables, as appropriate. Kaplan–Meier analysis was used to estimate the one-year survival and one-year PFS. To identify the risk factors for one-year survival, Cox-regression analysis was performed. Furthermore, to mitigate selection bias and potential confounders by equating baseline variables between the Ate/Bev and TACE + RT groups, we employed a 2:1 nearest-neighbor propensity score matching (PSM) method. Two-sided *p*-values < 0.05 were considered significant. All statistical analyses were conducted using R software (version 4.3.0; http://carn.r-project.org accessed on 21 April 2023).

## 3. Results

### 3.1. Baseline Characteristics

The mean age of participants (N = 97) was 59.1 years, and 87 patients (89.7%) were male. The majority of patients (n = 73, 76.0%) had hepatitis B infection. The median alpha-fetoprotein (AFP) level was 240.3 ng/mL, and the mean tumor size was 8.2 cm. Approximately half of the included patients (n = 45, 46.4%) had grade-4 PVTT (VP4) at the time of treatment, and there were no significant differences in baseline characteristics between the Ate/Bev and TACE + RT groups (Table 1).

### 3.2. One-Year Survival and PFS in the Entire Population

During the one-year follow-up, a total of 32 patients (33.0%), consisting of 5 patients from the Ate/Bev group and 27 patients from the TACE/RT group, died due to hepatic failure or HCC progression. The survival rate at one year was significantly higher in the Ate/Bev group compared to the TACE + RT group (79.7% vs. 50.3%, respectively; *p* = 0.041; Figure 2A). The one-year PFS was marginally higher in the Ate/Bev group (74.4% vs. 42.4%, respectively; *p* = 0.12; Figure 2B). Moreover, patients in the Ate/Bev group, who demonstrated a robust response at approximately 3 months, generally sustained this response throughout the first year.

### 3.3. Treatment Responses and AEs

During treatment, the Ate/Bev and TACE + RT showed a similar ORR (n = 15, 40.5% vs. n = 24, 40.0%, respectively; *p* = 1.000) and DCR (n = 28, 75.7% vs. n = 47, 78.3%, respectively; *p* = 0.957) (Figure 2C). Of the 97 patients, 16 patients (16.5%) experienced AEs, without significant differences between the Ate/Bev and TACE + RT groups (18.9% vs. 15.0%, respectively; *p* = 0.823). This lack of statistical difference persisted when we compared AEs grade-wise. Specifically, in the Ate/Bev group, the two most common AEs were proteinuria (n = 3; grade 1) and variceal bleeding (n = 2; grade 3). No immune-related AEs were observed during Ate/Bev therapy. In the TACE + RT group, variceal bleeding (n = 5; grade 3) was the most frequent AE (Table 2). Additionally, two patients in the TACE + RT group experienced HCC rupture (grade 4).

### 3.4. Subgroup Analysis

Because most patients had extensive HCC burden (n = 86, 88.7%), we further evaluated the efficacy of both treatments in those patients. Extensive HCC burden was defined as HCC patients with multiple tumors, a tumor size of ≥7 cm, or VP4 PVTT. Due to the poor prognosis associated with VP4 PVTT compared to VP1–3 PVTT, patients with VP4 were included in the extensive-HCC-burden category [14,17]. In patients with extensive HCC burden, the Ate/Bev group displayed a higher one-year survival rate compared to the TACE + RT group (85.7% vs. 46.2%, respectively; *p* = 0.005; Figure 3A). The one-year PFS was marginally higher in the Ate/Bev group compared to the TACE + RT group (75.4% vs. 40.8%, respectively; *p* = 0.088; Figure 3B). A superior one-year survival rate and marginally higher one-year PFS in the Ate/Bev group compared to the TACE + RT group were also demonstrated in patients with multiple intrahepatic HCC (n = 68, 70.1%; Appendix A).

To assess the potential efficacy of TACE in patients with unilobar intrahepatic HCC, we analyzed treatment outcomes in this patient group (n = 63, 64.9%). The one-year survival was similar between the Ate/Bev and TACE + RT groups (70.2% vs. 52.8%, respectively; *p* = 0.37; Figure 3C). The Ate/Bev group also showed a one-year PFS comparable to that of the TACE + RT group (*p* = 0.36, Figure 3D). Comparable results were consistently observed in patients with unilobar intrahepatic HCC without VP4 PVTT (Appendix A), suggesting the possibility of TACE + RT as an alternative treatment option in these patient groups.

### 3.5. Factors Associated with One-Year Survival and PSM Analysis

Given the overall superior efficacy of the Ate/Bev group compared to the TACE + RT group, we sought to reaffirm the effectiveness of Ate/Bev treatment using Cox-regression analysis and PSM. In Cox-regression analysis, the Ate/Bev treatment was documented as the only significant risk factor for one-year survival (hazard ratio, 0.38; 95% confidence interval, 0.15–1.00; *p* = 0.049; Table 3).

Finally, we compared the treatment outcomes of the two groups after PSM to further diminish any selection bias and possible confounders. The baseline characteristics were well balanced between the Ate/Bev (n = 27) and TACE + RT (n = 32) groups after PSM (Table 1). In the PSM model, the Ate/Bev group significantly outperformed the TACE + RT group in terms of both one-year survival (*p* = 0.02; Figure 4A) and one-year PFS (*p* = 0.01; Figure 4B), reaffirming the superior efficacy of the Ate/Bev treatment compared to TACE + RT.

## 4. Discussion

This multicenter cohort study is the first to compare the clinical outcomes of Ate/Bev and TACE + RT therapies in patients with PVTT who had no metastasis. Through detailed analyses, our study revealed that Ate/Bev treatment displayed better one-year survival compared to TACE + RT treatment. Furthermore, in the PSM analysis, both one-year survival and PFS rates were higher in the Ate/Bev treatment group. Thus, these results suggest that Ate/Bev treatment may be advantageous for the treatment of patients with PVTT.

Although Ate/Bev treatment has become a first-line therapy in advanced HCC cases, there are several other treatment options for patients with PVTT, including TACE + RT and hepatic artery infusion chemotherapy [11,18,19]. In addition, transarterial radioembolization presents an intriguing option, offering internal radiation without inducing vessel occlusion [20]. Among these treatments, as the TACE + RT has demonstrated superior outcomes compared to sorafenib [10], our study results elucidate the optimal choice for treating patients with PVTT by comparing Ate/Bev and TACE + RT. Indeed, the superior effectiveness of Ate/Bev therapy in our study reinforces the use of Ate/Bev treatment as the standard treatment for advanced HCC. In alignment with findings from the study by Richard Finn et al. [4], patients in the Ate/Bev group who responded well early on appeared to maintain this response without progression throughout the study period, potentially contributing to the superior outcomes of Ate/Bev therapy. Moreover, our results for one-year survival and PFS appear slightly superior to those reported in previous clinical trials [4,17]. This difference might be due to the distinct patient populations between the studies. Specifically, our study focused on HCC patients with PVTT who had no distant metastasis, and we included patients both with and without VP4 PVTT. These characteristics may have contributed to the more favorable outcomes observed in our cohort. Through this comparison, our findings underscore the potential advantages of Ate/Bev therapy over TACE + RT in treating HCC patients with PVTT.

Along with our novel results, suggesting the superior effectiveness of the Ate/Bev group, we also performed subgroup analyses to identify the preferred groups for each treatment. The superior outcome of the Ate/Bev therapy was constantly observed in patients with extensive HCC burden in our study. These results might be attributable to difficulties in embolizing feeding vessels during TACE in patients with VP4 PVTT and to the diminished efficacy of TACE in patients with a tumor size of ≥7 cm [21,22]. Meanwhile, patients with unilobar disease demonstrated comparable outcomes between the two groups. Given that TACE is technically more feasible for treating unilobar disease compared to bilobar disease, particularly in patients without VP4 PVTT [11,22], it seems that in addition to the Ate/Bev treatment, TACE + RT treatment could be considered as an alternative treatment option for these patient groups.

When treating HCC patients with PVTT, it is crucial to be mindful of potential AEs. In our analysis, we did not observe significant differences in the occurrence of AEs between the Ate/Bev and TACE + RT groups. However, grade 3 or 4 AEs, such as variceal bleeding or HCC rupture, were somewhat more prevalent in the TACE + RT group, indicating a potentially higher risk of severe toxicities with this approach. These observations bolster the safety profile of Ate/Bev therapy for HCC patients with PVTT. Meanwhile, when administering Ate/Be therapy, one must remain vigilant about the potential risk of reactivation of pre-existing autoimmune disease [20], despite the absence of patients with such diseases in our cohort.

Our study highlights the efficacy of Ate/Bev treatment, even in patients with PVTT. Atezolizumab, which inhibits PD-L1 on tumor cells or tumor-infiltrating immune cells, recovers the function of effector CD8+ T cells [23,24,25]. Moreover, bevacizumab shows synergistic effects by vessel normalization and a reduction in the immunomodulatory effect of VEGF on immune cells [26,27,28]. Considering the therapeutic mechanism of RT, which induces DNA damage and the remodeling of the tumor microenvironment, future studies using a combination of RT and Ate/Bev are necessary to improve the prognosis of patients with PVTT [29,30].

Our study has several limitations. First, there is the retrospective design of the present study. Second, this study analyzed a relatively small number of patients and had a short follow-up period. Because the Ate/Bev treatment was only covered by the Korean National Health Insurance starting from May 2022, its use has only recently become more widespread, resulting in a relatively short follow-up duration in our study. The one-year survival and PFS duration might seem short, but it is reasonable to use them to gauge the efficacy of both treatments, given the unfavorable prognosis in HCC patients with PVTT. Notably, this is the first study to compare the outcomes of the two treatments in HCC patients with PVTT. Despite the limited number of patients included, this was a multicenter study that reflected real-world data. Through detailed analyses, including PSM, our study provides insights into the superior efficacy of Ate/Bev therapy compared to TACE + RT therapy in treating HCC patients with PVTT.

## 5. Conclusions

In conclusion, our study suggests that Ate/Bev treatment yields superior clinical outcomes compared to TACE + RT in HCC patients with PVTT. In the context of current Korean HCC guidelines, which recommend both treatment options, our results could influence future updates by advocating for Ate/Bev as the primary treatment option for this patient population. For clinicians following different guidelines or practices, our findings offer evidence for considering Ate/Bev as a front-line therapy for HCC with PVTT. With respect to TACE + RT, it could also serve as an alternative treatment strategy alongside Ate/Bev therapy in patients with unilobar intrahepatic HCC. Given these observations, larger studies are warranted to further validate our findings.

## Figures and Tables

**Figure 1 cancers-15-04423-f001:**
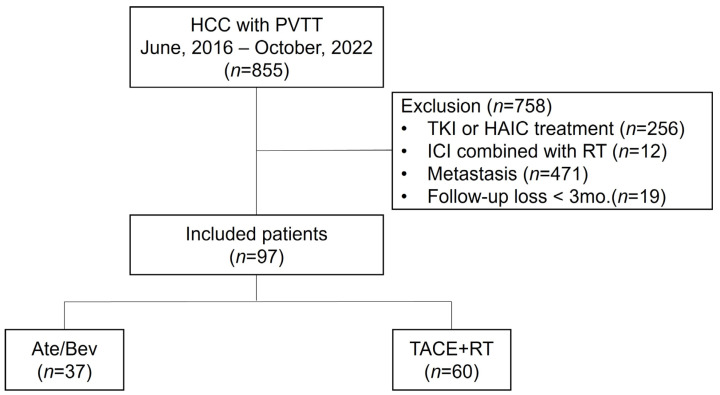
Study flow chart. HCC—hepatocellular carcinoma; VP4 PVTT—major portal vein tumor thrombus; TKIs—tyrosine kinase inhibitors; HAIC—hepatic artery infusion chemotherapy; ICIs—immune checkpoint inhibitors; RT—radiotherapy.

**Figure 2 cancers-15-04423-f002:**
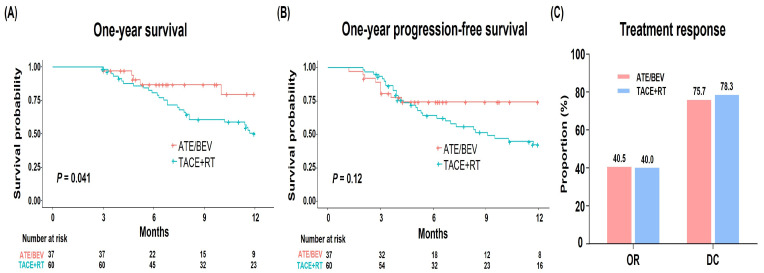
(**A**) One-year survival, (**B**) progression-free survival, and (**C**) treatment response in the entire cohort.

**Figure 3 cancers-15-04423-f003:**
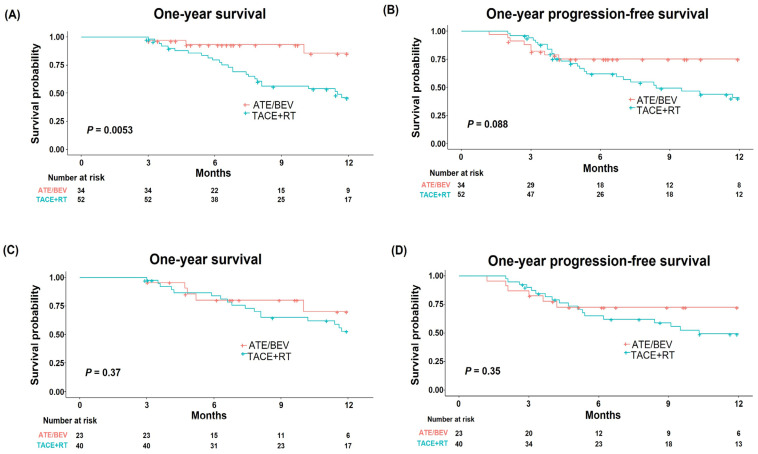
One-year survival and progression-free survival in patients with (**A**,**B**) extensive HCC burden and (**C**,**D**) unilobar intrahepatic HCC. Extensive HCC burden was defined as HCC patients with multiple tumors, a tumor size of ≥7 cm, or VP4 PVTT. HCC—hepatocellular carcinoma; VP4 PVTT—major portal vein tumor thrombus.

**Figure 4 cancers-15-04423-f004:**
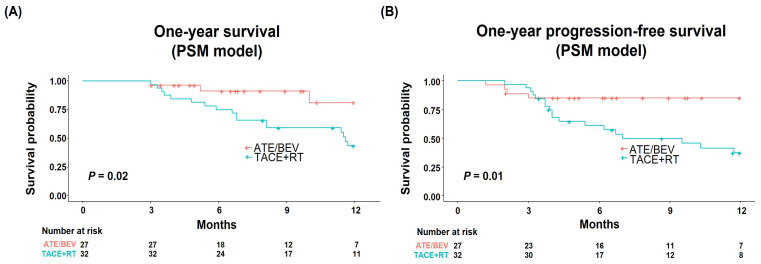
(**A**) One-year survival and (**B**) progression-free survival after propensity score matching.

**Table 1 cancers-15-04423-t001:** Baseline characteristics of entire population before and after propensity score matching.

Variables	Total (n = 97)	General Model			Propensity Score Matching Model		
		Ate/Bev Group	TACE + RT Group	*p*-Value	Ate/Bev Group	TACE + RT Group	*p*-Value
		(n = 37)	(n = 60)		(n = 27)	(n = 32)	
Age, years	59.1 ± 10.7	60.7 ± 12.9	58.1 ± 9.0	0.283	58.1 ± 12.0	59.6 ± 9.1	0.578
Male	87 (89.7%)	35 (94.6%)	52 (86.7%)	0.366	25 (92.6%)	30 (93.8%)	1.000
Cause (n, %)				0.074			0.954
HBV	73 (76.0%)	24 (64.9%)	49 (81.7%)		20 (74.1%)	25 (78.1%)	
Others	23 (24.2%)	13 (35.1%)	11 (18.3%)		7 (25.9%)	7 (21.9%)	
Treatment-naïve (n, %)	67 (69.1%)	22 (59.5%)	45 (75.0%)	0.167	16 (59.3%)	25 (78.1%)	0.199
AST, IU/mL	70.5 ± 46.1	64.1 ± 35.6	74.5 ± 51.5	0.245	68.5 ± 38.2	78.5 ± 56.6	0.425
ALT, IU/mL	42.2 ± 31.0	35.0 ± 23.7	46.5 ± 33.6	0.051	38.6 ± 26.6	49.2 ± 41.3	0.240
T.bil, mg/dL	1.0 ± 0.6	0.9 ± 0.7	1.1 ± 0.5	0.205	1.0 ± 0.7	1.1 ± 0.5	0.945
Alb, g/dL	3.7 ± 0.6	3.8 ± 0.4	3.7 ± 0.7	0.158	3.9 ± 0.4	3.9 ± 0.4	0.855
PLT, 103/μL	151.7 ± 64.2	162.6 ± 66.0	145.0 ± 62.6	0.190	169.0 ± 69.2	151.9 ± 66.6	0.339
INR.	1.1 [1.0; 1.2]	1.1 [1.0; 1.1]	1.1 [1.0; 1.2]	0.293	1.1 [1.0; 1.2]	1.1 [1.0; 1.2]	0.976
Cr, mg/dL	0.8 [0.7; 0.9]	0.7 [0.7; 0.9]	0.8 [0.6; 0.9]	0.442	0.7 [0.7; 0.9]	0.9 [0.6; 1.0]	0.488
CP class A	84 (86.6%)	34 (91.9%)	50 (83.3%)	0.371	24 (88.9%)	27 (84.4%)	0.902
AFP, ng/mL	240.3 [23.9; 1917.6]	484.4 [40.0; 3175.0]	189.2 [20.4; 1408.4]	0.277	237.0 [29.9; 2730.0]	274.6 [27.1; 1408.4]	0.632
PIVKA, mAU/mL	1723.0 [161.0; 6670.0]	2877.0 [230.0; 7521.0]	1344.0 [140.9; 5870.5]	0.653	2877.0 [245.4; 7753.3]	3436.3 [208.8; 7065.6]	0.849
Tumor Size, cm	8.2 ± 4.6	7.6 ± 4.8	8.5 ± 4.5	0.354	7.8 ± 5.0	8.8 ± 4.9	0.448
Multiple intrahepatic HCC (≥2 nodule)	68 (70.1%)	27 (73.0%)	41 (68.3%)	0.798	19 (70.4%)	23 (71.9%)	1.000
Bilobar intrahepatic HCC	34 (35.1%)	14 (37.8%)	20 (33.3%)	0.816	10 (37.0%)	12 (37.5%)	1.000
VP4 PVTT	45 (46.4%)	15 (40.5%)	30 (50.0%)	0.485	13 (48.1%)	15 (46.9%)	1.000

Ate/Bev—atezolizumab plus bevacizumab; TACE + RT—transarterial chemoembolization plus radiotherapy; AST—aspartate aminotransaminase; ALT—alanine aminotransaminase; T.bil—total bilirubin; Alb—albumin; PLT—platelet; INR—international normalized ratio; Cr, creatinine; CP—Child–Pugh; AFP—alpha-fetoprotein; PIVKA—Protein induced by vitamin K absence-II; HCC—hepatocellular carcinoma; VP4 PVTT—major portal vein tumor thrombus.

**Table 2 cancers-15-04423-t002:** Adverse events (AEs).

Adverse Events (AEs)	Total	Ate/Bev	TACE + RT	*p*-Value
	(n = 97)	(n = 37)	(n = 60)	
AE yes	16 (16.5%)	7 (18.9%)	9 (15.0%)	0.823
AE types according to grades				
Grade 1–2				0.068
- Proteinuria	3 (3.1%)	3 (8.1%)	0 (0.0%)	
- Others	2 (2.1%)	1 (2.7%)	1 (1.7%)	
Grade 3–4				0.524
- Varix bleeding	7 (7.2%)	2 (5.4%)	5 (8.3%)	
- Hepatic encephalopathy	2 (2.1%)	1 (2.7%)	1 (1.7%)	
- HCC rupture	2 (2.1%)	0 (0.0%)	2 (3.3%)	

**Table 3 cancers-15-04423-t003:** Cox-regression analysis for one-year survival.

Characteristics	HR	95% CI	*p*-Value
Age ≥ 60	1.75	0.86, 3.54	0.12
Male	2.15	0.51, 9.03	0.3
Ate/Bev Tx.	0.38	0.15, 1.00	0.049
AST ≥ 60, IU/mL	1.47	0.73, 2.96	0.3
ALT ≥ 40, IU/mL	1.16	0.58, 2.31	0.7
T.bil ≥ 1.0, mg/dL	1.97	0.98, 3.96	0.058
Alb ≤ 3.5, g/dL	1.13	0.46, 2.75	0.8
INR ≥ 1.1	1.00	0.50, 2.00	1.0
AFP ≥ 200, ng/mL	0.93	0.47, 1.87	0.8
PIVKA ≥ 40, mAU/mL	0.89	0.31, 2.55	0.8
Size ≥ 8.0, cm	1.42	0.71, 2.85	0.3
Multiple intrahepatic HCC	0.63	0.31, 1.28	0.2
Bilobar intrahepatic HCC	0.98	0.46, 2.07	1.0
VP4 PVTT	1.33	0.66, 2.66	0.4

HR—hazard ratio; CI—confidence interval; Ate/Bev Tx.; atezolizumab plus bevacizumab treatment; AST—aspartate transaminase, T.bil—total bilirubin; Alb—albumin; INR—international normalized ratio; AFP—alpha-fetoprotein; PIVKA—protein induced by vitamin K absence-II; HCC—hepatocellular carcinoma; VP4 PVTT—major portal vein tumor thrombus.

## Data Availability

Data are not available due to ethical issues.

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
