# Peer review of "A Real-World Comparative Analysis of Atezolizumab Plus Bevacizumab and Transarterial Chemoembolization Plus Radiotherapy in Hepatocellular Carcinoma Patients with Portal Vein Tumor Thrombosis"

_cancers, 2023, doi:10.3390/cancers15174423_

Round 1

Reviewer 1 Report

The study is interesting and overall well written. The authors should explain why they used the propensity score matching considering that the sample size did not change significantly after matching (so very few patients were excluded after matching).

In the West, TACE is not used in the case of portal vein thrombosis due to the high risk of complications. The authors should comment on this aspect and mention that there also are other loco-regional therapies in this case such as radioembolization (cite the recent MA: PMID: 27366304 )

THe authors should comment on the safety of these drugs (ICIs) with particular focus on the risk of re-activation of pre-existing autoimmune diseases (in this regard, cite the recent MA: PMID: 33314269)

Author Response

Point-by-Point Response to Reviewer I

Q1) The study is interesting and overall well written. The authors should explain why they used the propensity score matching considering that the sample size did not change significantly after matching (so very few patients were excluded after matching).

Ans) We appreciate the reviewer’s insightful comment. Given the retrospective nature of our study design, we implemented propensity score matching (PSM) analysis to minimize selection bias by equating baseline variables between the Ate/Bev and TACE+RT groups. Through PSM analysis, we further solidified the evidence supporting the superior efficacy of the Ate/Bev over TACE+RT therapy in both one-year survival and progression-free survival (PFS) rates. Following the reviewer’s comment, we have expanded on the rationale behind using the PSM analysis in the revised manuscript as follows:

“Furthermore, to mitigate selection bias and potential confounders by equating baseline variables between the Ate/Bev and TACE+RT groups, we employed a 2:1 nearest-neighbor propensity score matching (PSM) method.” (On page 4)

“Finally, we compared the treatment outcomes of the two groups were compared after PSM to further diminish any selection bias and possible confounders. The baseline characteristics were well-balanced between the Ate/Bev (n=27) and TACE+RT (n=32) groups after PSM (Table 1). In the PSM model, the Ate/Bev group significantly outperformed the TACE+RT group in terms of both one-year survival (P=0.02; Figure 4A) and one-year PFS (P =0.01; Figure 4B), reaffirming the superior efficacy of the Ate/Bev treatment compared to TACE+RT.” (On page 8)

Q2) In the West, TACE is not used in the case of portal vein thrombosis due to the high risk of complications. The authors should comment on this aspect and mention that there also are other loco-regional therapies in this case such as radioembolization (cite the recent MA: PMID: 27366304 )

Ans) We would like to thank the comment of the reviewer and totally agree with the opinions of the reviewer. We acknowledge that in Western practices, TACE might not be the primary choice for cases with portal vein tumor thrombosis (PVTT) due to the potential for complications. In our initial manuscript we mentioned transarterial radioembolization as an alternative treatment modality for PVTT on page 8.

To further clarify and enrich our manuscript based on the reviewer’s suggestion, we’ve enhanced the section on alternative treatments for HCC patients with PVTT as follows:

“Although Ate/Bev treatment has become a 1st line therapy in advanced HCC cases, there are several other treatment options for patients with PVTT, including TACE+RT and hepatic artery infusion chemotherapy [11,18,19]. In addition, transarterial radioembolization presents an intriguing option, offering internal radiation without inducing vessel occlusion [20]. Among these treatments, as the TACE+RT has demonstrated superior out-comes compared to sorafenib [10], our study results elucidate the optimal choice for treating patients with PVTT by comparing Ate/Bev and TACE+RT.” (On page 8-9)

(Please refer response Q1 of reviewer 2)

Q3) The authors should comment on the safety of these drugs (ICIs) with particular focus on the risk of re-activation of pre-existing autoimmune diseases (in this regard, cite the recent MA: PMID: 33314269)

Ans) We are grateful to the valuable comments of the reviewer and concur with the concerns raised. Indeed, the risk of reactivation of pre-existing autoimmune disease is a pertinent concern when evaluating the safety profile of ICI. In our cohort, we did not have any patients with pre-existing autoimmune disease, which may be attributed to the rarity of these conditions and the limited number of patients included in our study. Following the reviewer’s comment, we’ve incorporated remarks on the safety of ICIs, particularly emphasizing the risk of re-activation of autoimmune disease, in the revised manuscript:

“When treating of HCC patients with PVTT, it is crucial to be mindful of potential AEs. In our analysis, we did not observe significant differences in the occurrence of AEs between the Ate/Bev and TACE+RT groups. However, grade 3 or 4 AEs, such as variceal bleeding or HCC rupture, were somewhat more prevalent in the TACE+RT group, indicating a potentially higher risk of severe toxicities with this approach. These observations bolster the safety profile of Ate/Bev therapy for HCC patients with PVTT. Meanwhile, when administering Ate/Be therapy, one must remain vigilant about the potential risk of reactivation of pre-existing autoimmune disease [20], despite the absence of patients with such diseases in our cohort.” (On page 9)

Author Response

Point-by-Point Response to Reviewer II

This is a small retrospective study looking at systemic therapy (Atez/bev) vs. local therapies combination (TACE + radiation). It is a simple straightforward presentation that shows the benefit of using systemic therapy in HCC with PVTT, especially with high tumor burden. It re-affirms the known principle of treatment. The author may consider the following to make this a strong paper.

Q1) Please give us the current local guidelines (briefly) to manage HCC, because, for an international audience giving TACE in patients with PVTT may not be acceptable to start. This will give context to the study?

Ans) We appreciate the reviewer’s comment and understand the importance of contextualizing our study for an international audience. In our introduction, we referenced a randomized clinical trial (RCT) which compared the efficacy of TACE+RT to sorafenib in HCC patients with macrovascular invasion. The results from this study showed superior overall survival and progression-free survival with TACE+RT (as cited in JAMA Oncol 2018, 4, 661-669). Building on these findings, the Korean HCC guidelines recommend TACE+RT, along with Ate/Bev therapy, for HCC patients with macrovascular invasion, including PVTT. This recommendation is outlined in the “2022 KLCA-NCC Korea practice guidelines for the management of hepatocellular carcinoma” (Clin Mol Hepatol 2022 Vol. 28 Issue 4 Pages 583-705) as shown in the figure (below; Clin Mol Hepatol 2022 Vol. 28 Issue 4 Pages 583-705)

To address the reviewer’s request, we’ve expanded on this guideline in the revised manuscript as follows:

“Based on these results, the guidelines from 2022 Korean HCC guideline have placed TACE+RT (with a grade B for the quality of evidence) as a recommended treatment option for HCC patients with PVTT. This recommendation stands alongside Ate/Bev therapy, which holds a grade A for the quality of evidence [11].” (On page 2)

(Please refer response Q2 of reviewer 1)

Q2) Baseline characteristics: Consider adding previous or next line of therapies. As OS is the center of analysis, this information helps. If all of them are treatment naïve, please mention it.

Ans) We would like to thank the valuable comment of the reviewer. In our study, approximately 70% of the patients (n=67) were treatment naïve, with no significant differences observed between the Ate/Bev and TACE+RT groups. In response to the reviewer’s suggestion, we have added this information to Table 1 in the revised manuscript as follows:

“Table 1. Baseline characteristics of entire population before and after propensity score matching” (On page 4-5)

Q3) Baseline characteristics:  Multiple tumors: define the cut-off number. For some > 2 may be multiple and other its cab be > 5.

Ans) We are grateful to the reviewer’s insightful comments. In our study, we defined “multiple intrahepatic HCC” as the presence of two or more (≥ 2) nodules. Following the reviewer’s comment, we have included this cut-off number in the Table 1 as follows:

“Table 1. Baseline characteristics of entire population before and after propensity score matching” (On page 4-5)

Q4) Baseline characteristics: Can you mention few lines justifying PVTT in the main portal vein should be considered high-tumor burden?

Ans) We appreciate the comments of the reviewer. In our study, patients with PVTT in the main portal vein (VP4) were classified as having extensive HCC burden due to their poor prognosis compared to those with PVTT in branches of the portal vein (VP1-3), as reported in previous studies (J Hepatocellular Carcinoma 2021;8:1089-1115 and JHEP 2022;76:862-873). Indeed, Cheng et. al. classified VP4 in the high-risk group evaluating the efficacy of safety of Ate/Bev therapy from IMbrave150 (JHEP 2022;76:862-873). Following the reviewer’s comment, we have revised the manuscript to include the following:

“Extensive HCC burden was defined as HCC patients with multiple tumors, a tumor size of ≥ 7cm or VP4 PVTT. Due to the poor prognosis associated with VP4 PVTT compared to VP1-3 PVTT, patients with VP4 were included in the extensive HCC burden category [14,17]. In patients with extensive HCC burden, the Ate/Bev group displayed a higher one-year survival rate compared to the TACE+RT group (85.7% vs. 46.2%, respectively; P=0.005; Figure 3A).” (On page 6)

Q5) Results: The primary outcome was one-year survival. It should be before response rates in the

results.

Ans) We are grateful to the reviewer’s observation and agree that the primary outcome should take precedence in the Results section. In line with the reviewer’s recommendation, we have rearranged the Results section to present the on-year survival data before discussing the response rates. This changes has been implemented in the revised manuscript as follows:

“3.2. One-year survival and PFS in the entire population

During the one-year follow-up, a total of 32 patients (33.0%), consisting of 5 patients from the Ate/Bev group and 27 patients from the TACE/RT group, died due to hepatic failure or HCC progression. Survival rate at one-year was significantly higher in the Ate/Bev group compared to the TACE+RT group (79.7% vs. 50.3 %, respectively; P=0.041; Figure 2A). The one-year PFS were marginally higher in the Ate/Bev group (74.4% vs. 42.4%, respectively; P=0.12; Figure 2B). Moreover, patients in the Ate/Bev group, who demonstrated a robust response at approximately 3 months, generally sustain this response throughout the first year.

3.3. Treatment responses and AEs

During treatment, the Ate/Bev and TACE+RT showed similar ORR (n=15, 40.5% vs. n=24 40.0%, respectively; P=1.000) and DCR (n=28, 75.7% vs. n=47, 78.3%, respectively; P=0.957) (Figure 2C). Of the 97 patients, 16 patients (16.5%) experienced AEs without significant differences between the Ate/Bev and TACE+ RT groups (18.9% vs. 15.0%, respectively; P=0.823). This lack of statistical difference persisted when we compared AEs grade-wise. Specifically, in the Ate/Bev group, the two most common AEs were proteinuria (n=3; grade 1) and variceal bleeding (n=2; grade 3). No immune-related AEs were observed during Ate/Bev therapy. In the TACE+RT group, variceal bleeding (n=5; grade 3) was the most frequent AEs (Table 2). Additionally, two patients in the TACE+RT group experienced HCC rupture (grade 4).” (On page 5-6)

Q6) The figures are not legible. Please add bigger ones (all of them).

Ans) We appreciate the reviewer’s feedback regarding the legibility of the figures. In response, we have enlarged all the figures to improve readability in the revised manuscript.

Q7) Consider mentioning AE-s, grade-wise G1-2 vs G3-4 vs G5? Use any CTCAE. Mention few sentences of occurrence or no immune-related adverse events.

Ans) We would like to thank the reviewer’s valuable comments on AEs. In our study, AEs were evaluated and graded according to the CTCAE 5.0. We did not observe any statistically significant differences in AEs between the Ate/Bev and TACE+RT groups, either for grades G1-2 or G3-4. However, it is worth noting that HCC rupture, categorized as a grade 4 AE, was only observed in the TACE+RT group. In light of the reviewer’s suggestion, we have revised the manuscript as follows:

“Of the 97 patients, 16 patients (16.5%) experienced AEs without significant differences between the Ate/Bev and TACE+ RT groups (18.9% vs. 15.0%, respectively; P=0.823). This lack of statistical difference persisted when we compared AEs grade-wise. Specifically, in the Ate/Bev group, the two most common AEs were proteinuria (n=3; grade 1) and variceal bleeding (n=2; grade 3). No immune-related AEs were observed during Ate/Bev therapy. In the TACE+RT group, variceal bleeding (n=5; grade 3) was the most frequent AEs (Table 2). Additionally, two patients in the TACE+RT group experienced HCC rupture (grade 4).” (On page 6)

“Table 2. Adverse events (AEs)”

Q8) Conclusion: for international audiences, it is not news but we do not understand the context of

this stud’s importance. Please address this. What is the current practice and how will this study

re-affirm it.

Ans) We appreciate the reviewer’s comment. Following the reviewer’s comment, we added the comments about the current Korean guideline and the impacts of our study to the clinical practice as well as the treatment guideline. We have made the revisions to the manuscript as follows:  

“In conclusion, our study suggests that Ate/Bev treatment yields superior clinical outcomes compared to TACE+RT in HCC patients with PVTT. In the context of current Korean HCC guidelines, which recommend both treatment options, our results could influence future updates by advocating Ate/Bev as the primary treatment option for this patient population. For clinicians following different guidelines or practices, our findings offer evidence to consider Ate/Bev as a front-line therapy for HCC with PVTT. With respect to TACE+RT, this could also serve as an alternative treatment strategy alongside Ate/Bev therapy in patients with unilobar intrahepatic HCC. Given these observations, larger studies are warranted to further validate our findings.” (On page 10)

Reviewer 3 Report

Good paper deserving publication.

Some minor comments to solve before :

1- multicentric trial: will it be possible to say (if right) "in 8 University Hospitals part of the Catholic University in Korea" ?

2-  chapter 2.2 : please precise how you define « patients with appropriate portal blood flow », by definition in most cases of PVTT the portal flow is at least decrease, particularly in VP4 ; does it mean that in many cases patients had only a « Lipiodolization » without real embolization ? to better describe 

3- chapter 2.3 : assessment of outcome : was it so easy to precisely measure tumors in patients with PVTT treated by TACE then external beam radiotherapy ?

4- chapter 3.2 : in the TACE+RT group were observed as SAE, 5 variceal bleedings and 2 tumor ruptures without any death ; grading / evaluating these SAE from Grade 1 to grade 4 (5 ?) is mandatory;

5- Discussion : another advantage of Atezo-Bev treatment is safety with, by contrast, major toxicities in  patients treated by TACE + radiotherapy;

6- Discussion: about the lenght of tumor response, in the seminal trial by Richard Finn et al., it is stated that the estimated percentage of patients with duration of response longer than 6 months was 87.6% in the Atezo-Bev group; it is not so intriguing.

7- Discussion: I think that taking into account the prognosis, the short follow-up is not really a major limitation for this study. The low number of included patients is a limitation.

8- Discussion: In my mind it would be of interest to elaborate a little bit more on the survival you observed, because in the IMBrave 150 study, including less than 40% patients with PVTT the 1 year OS and 1 year PFS were respectively of 67.2% and under 40% with a lower median OS for patients with PVTT (A-L Cheng, J Hepatol 2022) that is to say by far belowyour data (and B virus infection was not a good factor); could you comment: selection bias in both arms? 

Author Response

Point-by-Point Response to Reviewer III

Q1) multicentric trial: will it be possible to say (if right) "in 8 University Hospitals part of the Catholic University in Korea" ?

Ans) We appreciate the reviewer’s insightful feedback. Following the reviewer’s comment, we revised manuscript as follows:

“In this retrospective, multicenter study conducted across eight university hospitals part of The Catholic University of Korea, HCC patients with PVTT (n=855), who were treated between June 2016 and October 2022, were consecutively screened for eligibility.” (On page 2)

Q2) chapter 2.2 : please precise how you define « patients with appropriate portal blood flow », by definition in most cases of PVTT the portal flow is at least decrease, particularly in VP4 ; does it mean that in many cases patients had only a « Lipiodolization » without real embolization ? to better describe

Ans) We are grateful to the comments of the reviewer. As highlighted by the reviewer, our procedure involved performing embolization using a gelatin sponge for patients with adequate portal blood flow. Conversely, we abstained from embolization in patients exhibiting severe impairment in portal blood flow. Following the reviewer’s feedback, we have revised the manuscript as follows:

“Subsequently, the feeding arteries were embolized using a mixture of 5-10 mL of cisplatin and iodized oil (Lipiodol Ultra-Fluide; Laboratoire Guerbet), followed by a gelatin sponge (Gelfoam; Upjohn) in patients with appropriate portal blood flow. However, for patients with severely impaired portal blood flow, the embolization was not omitted. TACE treatment was repeated every 6-8 weeks according to the patient’s status [10].” (On page 3)

Q3) chapter 2.3 : assessment of outcome : was it so easy to precisely measure tumors in patients with PVTT treated by TACE then external beam radiotherapy ?

Ans) We would like to thank the valuable comments of the reviewer. In our study, the assessment of tumor measurements and treatment responses was conducted by two expert radiologists. These radiologists were blinded to both the treatment modalities and the clinical outcomes of the patients. Following the reviewer’s comment, we have made the revisions to the manuscript as follows:  

“Treatment responses were assessed by multiphase liver CT or MRI every 6 to 9 weeks according to Response Evaluation Criteria in Solid Tumors 1.1 (RECIST 1.1) [15]. Measurements of the tumor and responses to the treatments were independently reviewed by two expert radiologists (DJ Shim and D Kim) without knowledge of the treatment methods and clinical outcomes.” (On page 3-4)

Q4) chapter 3.2 : in the TACE+RT group were observed as SAE, 5 variceal bleedings and 2 tumor ruptures without any death ; grading / evaluating these SAE from Grade 1 to grade 4 (5 ?) is mandatory;

Ans) We appreciate the reviewer’s comment and totally agree with the reviewer’s comment. Following the reviewer’s comment, we have incorporated grading for AEs in the revised manuscript as follows:

“Of the 97 patients, 16 patients (16.5%) experienced AEs without significant differences between the Ate/Bev and TACE+ RT groups (18.9% vs. 15.0%, respectively; P=0.823). This lack of statistical difference persisted when we compared AEs grade-wise. Specifically, in the Ate/Bev group, the two most common AEs were proteinuria (n=3; grade 1) and variceal bleeding (n=2; grade 3). No immune-related AEs were observed during Ate/Bev therapy. In the TACE+RT group, variceal bleeding (n=5; grade 3) was the most frequent AEs (Table 2). Additionally, two patients in the TACE+RT group experienced HCC rupture (grade 4).” (On page 6)

“Table 2. Adverse events (AEs)”

Q5) Discussion : another advantage of Atezo-Bev treatment is safety with, by contrast, major toxicities in  patients treated by TACE + radiotherapy;

Ans) We are grateful to the insightful comment of the reviewer and totally agree with the opinions of the reviewer. Although there were no statistical differences in AEs between the Ate/Bev and the TACE+RT groups, grade 3 or 4 AEs, including variceal bleeding or HCC rupture, were marginally more prevalent in the TACE+RT group. This result suggests the potentially higher risk of severe toxicities in TACE+RT therapy compared to Ate/Bev treatment. Following the reviewer’s comment, we added discussion about AEs in Ate/Bev and TACE+RT therapy as follows:

“When treating of HCC patients with PVTT, it is crucial to be mindful of potential AEs. In our analysis, we did not observe significant differences in the occurrence of AEs be-tween the Ate/Bev and TACE+RT groups. However, grade 3 or 4 AEs, such as variceal bleeding or HCC rupture, were somewhat more prevalent in the TACE+RT group, indicating a potentially higher risk of severe toxicities with this approach. These observations bolster the safety profile of Ate/Bev therapy for HCC patients with PVTT. Meanwhile, when administering Ate/Be therapy, one must remain vigilant about the potential risk of reactivation of pre-existing autoimmune disease [20], despite the absence of patients with such diseases in our cohort.” (On page 9)

Q6) Discussion: about the length of tumor response, in the seminal trial by Richard Finn et al., it is stated that the estimated percentage of patients with duration of response longer than 6 months was 87.6% in the Atezo-Bev group; it is not so intriguing.

Ans) We appreciate the comments of the reviewer. Following the reviewer’s comment, we rephrased the manuscript as follows:

“Moreover, patients in the Ate/Bev group, who demonstrated a robust response at approximately 3 months, generally sustain this response throughout the first year.” (On page 5)

“In alignment with findings from the study by Richard Finn et, al. [4], patients in the Ate/Bev group who responded well early on appeared to maintain this response without progression throughout the study period, potentially contributing to the superior outcomes of Ate/Bev therapy.” (On page 9)

Q7) Discussion: I think that taking into account the prognosis, the short follow-up is not really a major limitation for this study. The low number of included patients is a limitation.

Ans) We would like to thank the valuable comments of the reviewer. As highlighted by the reviewer, the number of included patients was a limitation of our study, while the short follow-up is acceptable considering a poor prognosis of HCC patients with PVTT. Following the reviewer’s comment, we revised manuscript as follows:

“The one-year survival and PFS duration might seem short, but it’s reasonable to use them to gauge the efficacy of both treatments, given the unfavorable prognosis in HCC patients with PVTT. Notably, this is the first study to compare the outcomes of the two treatments in HCC patients with PVTT. Despite the limited number of patients included, this was multicenter study that reflected real-world data. Through detailed analyses, including PSM, our study provides insights into the superior efficacy of Ate/Bev therapy compared to TACE+RT therapy in treating HCC patients with PVTT.” (On page 9-10)

Q8) Discussion: In my mind it would be of interest to elaborate a little bit more on the survival you observed, because in the IMBrave 150 study, including less than 40% patients with PVTT the 1 year OS and 1 year PFS were respectively of 67.2% and under 40% with a lower median OS for patients with PVTT (A-L Cheng, J Hepatol 2022) that is to say by far below your data (and B virus infection was not a good factor); could you comment: selection bias in both arms?

Ans) We are grateful to the insightful comments of the reviewer. This discrepancy might stem from the differences in the characteristics of included patients between trials and our study. Our study included HCC patients having PVTT without distant metastasis. Moreover, we included patients both with and without VP4 PVTT. It might contribute to the superior results of our study. Following the reviewer’s comment, we added the explanation of these differences between trials and our results in the revised manuscript as follows:

“In alignment with findings from the study by Richard Finn et, al. [4], patients in the Ate/Bev group who responded well early on appeared to maintain this response without progression throughout the study period, potentially contributing to the superior out-comes of Ate/Bev therapy. Moreover, our results for 1-year survival and PFS appear slightly superior to those report-ed in previous clinical trials [4,17]. This difference might be due to the distinct patient populations between the studies. Specifically, our study focused on HCC patients with PVTT who had no distant metastasis, and we included patients both with and without VP4 PVTT. These characteristics may have contributed to the more favorable outcomes observed in our cohort. Through this comparison, our findings underscore the potential advantages of Ate/Bev therapy over TACE+RT for treating HCC patients with PVTT.” (On page 9)

Round 2

Reviewer 1 Report

The revised version of the paper is OK. Thank you!